# Effect of Recycled Iron Powder as Fine Aggregate on the Mechanical, Durability, and High Temperature Behavior of Mortars

**DOI:** 10.3390/ma13051168

**Published:** 2020-03-05

**Authors:** Md Jihad Miah, Md Kawsar Ali, Suvash Chandra Paul, Adewumi John Babafemi, Sih Ying Kong, Branko Šavija

**Affiliations:** 1Department of Civil Engineering, University of Asia Pacific, Dhaka 1205, Bangladesh; jihad.miah@uap-bd.edu (M.J.M.); kawsar.uap@gmail.com (M.K.A.); 2Department of Civil Engineering, International University of Business Agriculture and Technology, Dhaka 1230, Bangladesh; suvashpl@iubat.edu; 3Department of Civil Engineering, Stellenbosch University, Private Bag X1, Matieland, Stellenbosch 7602, South Africa; ajbabafemi@sun.ac.za; 4Discipline of Civil Engineering, Monash University Malaysia, Subang Jaya, Selangor 47500, Malaysia; kong.sih.ying@monash.edu; 5Microlab, Faculty of Civil Engineering and Geosciences, Delft University of Technology, 2628CN Delft, The Netherlands

**Keywords:** recycled iron powder, mortar, mechanical properties, durability, temperature resistance

## Abstract

This study evaluates the mechanical, durability, and residual compressive strength (after being exposed to 20, 120, 250, 400 and 600 °C) of mortar that uses recycled iron powder (RIP) as a fine aggregate. Within this context, mechanical strength, shrinkage, durability, and residual strength tests were performed on mortar made with seven different percentages (0%, 5%, 10%, 15%, 20%, 30% and 50%) of replacement of natural sand (NS) by RIP. It was found that the mechanical strength of mortar increased when replaced with up to 30% NS by RIP. In addition, the increase was 30% for compressive, 18% for tensile, and 47% for flexural strength at 28 days, respectively, compared to the reference mortar (mortar made with 100% NS). Shrinkage was observed for the mortar made with 100% NS, while both shrinkage and expansion occurred in the mortar made with RIP, especially for RIP higher than 5%. Furthermore, significantly lower porosity and capillary water absorption were observed for mortar made with up to 30% RIP, compared to that made with 100% NS, which decreased by 36% for porosity and 48% for water absorption. As the temperature increased, the strength decreased for all mixes, and the drop was more pronounced for the temperatures above 250 °C and 50% RIP. This study demonstrates that up to 30% RIP can be utilized as a fine aggregate in mortar due to its better mechanical and durability performances.

## 1. Introduction

Aggregates are essential in concrete production. Usually inert, they are used to minimize costs, lessen shrinkage and creep, and improve the overall mechanical strength of concrete [1]. Therefore, ensuring the availability of good quality aggregates is essential for concrete construction. In addition, the material should be cheap, sustainable, and environmentally-friendly [2,3]. However, in the Netherlands, for example, the availability of river aggregates is decreasing due to high demand, leading to an increased need for alternative sources, such as the use of concrete waste as an aggregate [4]. On the other hand, some islands have no access to natural aggregates and are looking into using coral reef aggregates from the sea in concrete production [5]. 

In general, coarse and fine aggregates occupy about 40–50% and 20–30%, respectively, of the total volume of hardened concrete. Natural sand (NS) is mainly used as a fine aggregate. The availability of natural sand, which is mostly extracted from riverbeds by dredging, is limited. Furthermore, the dredging process has a negative effect on the environment. Since the demand for concrete is increasing rapidly, especially in developing countries, it can be expected that depletion of natural fine aggregates will occur. As a result, it is essential to find possible alternative raw construction materials that could be used as fine aggregates in concrete construction works. Furthermore, considering the amount of aggregates used in concrete production in the world yearly (estimated at more than 10–11 billion tons 20 years ago), there is great potential for reuse of wastes as aggregate replacement [6,7,8]. 

Recycled iron powder (RIP) is the industrial byproduct of the grinding, cutting, and milling of finished iron products [9]. This byproduct is generated in enormous quantities by workshops, steel mills, and factories, in powder form [10,11]. Although reliable data on the quantity of RIP generated globally is lacking, it seems that there is an increasing rate of this material due to rapid urbanization around the world. According to Tayeh and Al Saffar [10], the production of iron powders contributes to around 5% of the total municipal waste. However, in the past, the shipments of iron powders were in the order of 200,000 tons in the U.S.A. and Canada, 120,000 tons in Europe, and 60,000 tons in Japan [11]. Furthermore, RIP can be hazardous to human health since it can be easily inhaled [12]. Studies have shown that RIP can potentially be used as a partial replacement of NS in concrete production. For example, Ghannam et al. [12] studied the effect of partial replacement of sand with the iron powder in concrete. Their study showed that up to 20% of sand could be replaced by iron powder without compromising the compressive and flexural strength of concrete. 

Similarly, Ismail and Al-Hashmi [13] observed an increase in compressive and flexural strength when 20% of sand was replaced with waste iron. An increase of 17% and 28% (compared to the reference concrete) was found for compressive and flexural strength, respectively. Other researchers reported that iron fillings can replace even higher percentages of sand without compromising the properties of concrete. Satyaprakash et al. [14] found that the compressive and splitting tensile strength of concrete made with 100% iron filings is about 26% higher than the concrete made with 100% natural sand. Furthermore, the replacement of NS with iron fillings resulted in significantly better abrasion resistance of concrete. Improvement of mechanical properties compared to normal concrete was also observed for concrete made with recycled scale and steel chips [15,16]. Other researchers found that the compressive strength of concrete containing iron filings was higher than the plain concrete. Besides, the presence of iron filings enhances the ductility of concrete [17]. Kumar et al. [18] investigated the effect of partial/total replacement of sand by iron ore tailing, as the fine aggregate on the compressive and flexural strength of reinforced concrete. The compressive strength was increased by up to 40% with the replacement of sand by iron ore tailing, while there was an enhancement of flexural strength for all percentages of sand replacement (10%, 20%, 30%, 40%, 60%, 80% and 100%). Singh and Siddique [19,20] studied the mechanical, microstructural, and durability performances of self-compacting concrete made with four different replacement percentages (0%, 10%, 25% and 40%) of natural fine aggregate by iron slag. Enhancement of mechanical strength (compressive, splitting tensile, and flexural) was observed for the concrete containing iron slag as the reference concrete. Furthermore, the durability of concrete made with iron slag was better than reference concrete due to dense microstructure of iron concrete.

On the other hand, Tayeh and Saffar [10] found that the incorporation of iron powder decreased the compressive strength of mortar due to a higher amount of voids, which may affect the compressive strength. In contrast, they observed a significant increase in flexural strength of mortar with the increased content of the iron powder. However, they also observed a reduction in workability with the increased percentage of iron powder due to heterogeneity and higher angularity of the waste iron powder. Other authors also reported that an increasing percentage of steel scale waste results in decreased compressive strength [21]. Therefore, the influence of the addition of RIP on the mechanical properties of concrete remains an open issue due to contradictory trends reported in the literature.

Most studies discussed above dealt with studying the effects of sand replaced by RIP on the mechanical properties of concrete. Mortar, on the other hand, contains no coarse aggregates; it is, therefore, expected that the effects of sand replacement by RIP on its properties will be more pronounced. Furthermore, almost no research has focused on the time-dependent properties of cementitious materials incorporating RIP, such as shrinkage. In addition, the durability of such materials and its resistance to high temperatures are not understood. This study, therefore, aims to elucidate the effects of RIP incorporation on the mechanical properties, shrinkage, durability, and high temperature resistance of mortar. An extensive investigation of the physical properties, fresh properties, hardened properties (i.e., compressive, tensile and flexural strength, shrinkage, and dry density), durability (porosity and water absorption), and decay of mechanical properties at high temperatures of mortar made with seven different percentages of replacement (0%, 5%, 10%, 15%, 20%, 30% and 50%) of NS by RIP were carried out. The results of this study provide a basis for further research on the practical applicability of mortar and concrete incorporating recycled iron powder.

## 2. Materials and Methods

### 2.1. Materials 

Natural sand (NS) and recycled iron powder (RIP) as fine an aggregate was used to study the mechanical and durability properties of mortar. Both NS and RIP were passed through a sieve opening of 4.75 mm (ASTM standard No. 4 sieve) to fall into the size range of a fine aggregate. The microscopic morphology of the RIP through Scanning Electron Microscopy (SEM, JSM 7600F, JEOL, Tokyo, Japan) was performed. The regular image of the RIP and SEM image are presented in Figure 1.

The specific gravity and absorption capacity of both fine aggregates (NS and RIP) were tested as per ASTM standards [22,23] (Table 1). The sieve analysis of the fine aggregates was conducted, and the results were compared with the upper and lower limits recommended in the ASTM C33 standard [24] (Figure 2). It was observed that both fine aggregates were within the limits of the ASTM C33 [24] standard curve. CEM II/A-M cement, which consists of 80–94% clinker, 6–20% of mineral admixture (slag and fly ash), and gypsum, was used as a binder for all mortar mixes. The physical properties of cement, such as normal consistency, initial and final setting time, and compressive strength at 28 days, were 27.5%, 110 min, 360 min, and 32 MPa, respectively.

The chemical composition of RIP and cement was determined by X-ray fluorescence (XRF, Lab Center XRF-1800, Shimadzu, Kyoto, Japan) analysis. The results of the XRF analyses are presented in Table 2. 

RIP contains about 8.46% SiO_2_, 87.46% Fe_2_O_3_, 0.87% Al_2_O_3_ and 1.08% CaO. A similar chemical composition was reported by Ghannam et al. [12]. As expected, the Fe_2_O_3_ content of RIP was significantly higher, resulting in higher specific gravity than that of natural sand (4.3 for RIP and 2.56 for NS, see Table 1).

### 2.2. Experimental Programs and Test Procedures

The mortar specimens were produced with a constant water to cement ratio (w/c) of 0.30, and superplasticizer (SikaPlast^®^-204 TH, Chonburi, Thailand), as the chemical admixture, was used (0.5% by mass of total cement) to ensure the workability of fresh mortar. The experimental program was divided into three parts: mechanical properties, durability (porosity and water absorption capacity), and high temperature tests. To investigate the effect of RIP on the mortar properties, seven different percentage replacements (0%, 5%, 10%, 15%, 20%, 30% and 50%) of natural sand (NS) by RIP were studied. The mixture proportions of the mortar mixes are summarized in Table 3. The workability of the mortar mixes was investigated by measuring the slump values of fresh mortar. In addition, the temperature of the fresh mortar mixes was measured.

Furthermore, the dimensional stability of the hardened mortar of all mixes was monitored according to the ASTM C 490 [25] using prismatic specimens (25 mm × 25 mm × 285 mm). A total of 21 specimens (3 specimens for each mix) were cast in steel molds in the laboratory to monitor the length change as a function of time. All the specimens were cured underwater (20 ± 2 °C) until the end of the test. As a reference, the first value was taken 24 h after casting. A high-accuracy digital dial gauge (JJ191268, FeelinGirl, China) with precision of 0.001 mm was used to monitor the length change of mortar.

#### 2.2.1. Mechanical Properties

Cube specimens (50 mm) for compression, briquet specimens for tension, and prism specimens (40 mm × 40 mm × 160 mm) for flexural strength tests were made and tested as per ASTM C109 [26], ASTM 307 [27], and ASTM 348 [28], respectively. The evolution of mechanical strength (compressive, tensile, and flexural) as a function of the age of mortar was investigated at 7, 14, 28 and 60 days. Moreover, the dry density of mortar specimens was measured on the same specimens that were used for the mechanical tests. All the mortar specimens were cast in steel molds in the laboratory. The specimens were demolded 24 h after casting and cured underwater (20 ± 2 °C) until the day of the tests.

#### 2.2.2. Porosity and Water Absorption

The porosity of the mortar mixes was investigated by employing a technique based on water absorption porosity according to the French standard NF P18-459 [29]. The tests were performed on a quarter-cylinder of mortar specimens, 104 mm in diameter and 50 mm thickness. Three specimens were used for each mortar mix, and the final porosity was calculated by the arithmetic mean of three specimens. The porosity of mortar (P, in %) was calculated as:(1)$$P=\frac{m_{s}-m_{d}}{m_{s}-m_{h}}$$
where *m_s_*, *m_d_* and *m_h_* are the masses of the saturated specimen, the dry specimen, and the hydrostatically measured mass, respectively. To perform the mass measurements, the specimens were first dried in an oven at a temperature of 105 ± 5 °C until a constant mass was reached (*m_d_*). Then, the specimens were water-saturated using a vacuum. A vacuum pressure of 25 mbar was attained and maintained for 4 h. After that, the specimens were left immersed in water for 24 h. Saturated specimens were used to measure the hydrostatic mass (*m_h_*). Finally, the saturated specimens were removed from the water and a damp cloth was used to remove excess surface water and attain a saturated surface dry (SSD) condition before measuring the saturated mass (*m_s_*).

The capillary water absorption capacity test was performed following the French recommendations AFPC-AFREM [30]. The water capillary absorption capacity tests were performed on disc specimens of 150 mm in diameter and 70 mm thickness. The mortar discs were cut very carefully from larger cylinders by a diamond blade saw, meant not to damage the specimen, which could allow faster migration of water from the bottom to the top. The upper and lower discs were discarded to avoid the edge effect which could influence the measurements. The curved surface of the specimen was sealed with epoxy to ensure unidirectional water flow (from bottom to top). Before testing, the specimens dried in an oven at a temperature of 80 °C until constant mass was reached. Oven-dried samples were placed vertically in water on average 3 mm diameter steel rods, to allow free access of water to the inflow. The mass of the specimens was recorded for up to 120 h of water absorption (until the stabilization of the mass of the specimens). At the beginning of the test, readings were collected more frequently (the lowest frequency of reading was every 15 min) and after that, at least one reading per hour was taken. The water level remained constant throughout the test. The absorption capacity was calculated from the mass in dry and wet conditions. Three specimens were used for each mortar mix to calculate the average water absorption capacity of the mortar.

#### 2.2.3. Residual Compressive Strength of Mortar After Exposure to High Temperature

The residual compressive strength of mortar was investigated on the cube (50 mm) specimens at room temperature after applying thermal loads of 120, 250, 400 and 600 °C at a slow heating rate of 2 °C/min. After reaching the target temperature, the temperature was stabilized for 28, 10, 4 and 4 h, respectively, for the temperatures of 120, 250, 400 and 600 °C to reach uniformity [31] in the mortar. The specimens were then cooled naturally by turning off the furnace and reaching room temperature inside the closed furnace. The reference specimens were tested at ambient temperature (20 °C). Three specimens were tested for each mixture and each high temperature exposure.

## 3. Results and Discussion

### 3.1. Fresh Properties of Mortar

The fresh properties of mortar made with seven different percentage replacements of NS by RIP were evaluated, by measuring the slump value and temperature of fresh mortar at the time of placing, and presented in Table 4. It was observed that mortars made with RIP had lower slump than the mortar made with NS. As the percentage replacements increased the slump value decreased. At 50% replacement the slump was 35% lower than the reference. This can be attributed to the highly angular and rough surface (see Figure 1b) of RIP, whereas the NS is relatively round. This resulted in the reduction of the flowability of the mortar made with RIP, due to more interlocking of the particles in the mix. Moreover, the reduction in the flowability could also be linked to the temperature of the fresh mortar at the time of placing. As shown in Table 4, as the temperature of the fresh mortar increased the workability of the mortar decreased, due to the higher heat of hydration of cement paste.

### 3.2. Hardened Properties of Mortar

#### 3.2.1. Dry Density

The evolution of the dry density of mortar was measured using the same mortar cubes used for compressive strength tests. The average dry density and normalized dry density of hardened mortar measured at 7, 14, 28 and 60 days are presented in Figure 3a. It was observed that the dry density of hardened mortar increased with an increase in the percentage replacement of NS by RIP (see Figure 3b). The average dry density of mortar at 28 days made with 0%, 5%, 10%, 15%, 20%, 30% and 50% RIP were 2233 kg/m^3^, 2337 kg/m^3^, 2377 kg/m^3^, 2445 kg/m^3^, 2497 kg/m^3^, 2541 kg/m^3^, and 2623 kg/m^3^, respectively. The maximum increase in dry density of mortar was observed when NS was replaced by 50% RIP, which was about 16–18% higher than that of the reference mortar for all curing ages (100% NS mortar, see Figure 3). The increase in the dry density can be attributed to the higher specific gravity (4.3 for RIP and 2.56 for NS) of RIP compared to NS. The higher density of mortar made with RIP is in agreement with the findings of other researchers [10,16].

#### 3.2.2. Compressive Strength

The compressive strength of mortar made with seven different percentage replacements (0%, 5%, 10%, 15%, 20%, 30% and 50%) of NS by RIP was conducted at 7, 14, 28 and 60 days. The results are given in Figure 4a. At up to 30% replacement of NS with RIP, the mortars had significantly higher strength compared to the reference mortar at all ages. Furthermore, although the compressive strength of the mortar with 50% RIP content was lower than of mortars with lower RIP percentages, it was higher than the reference mixture. Therefore, all mixtures with RIP addition had higher compressive strength compared to the reference at all ages. For example, the average compressive strengths of mortar containing 0%, 5%, 10%, 15%, 20%, 30% and 50% RIP measured at 28 days are 46.9, 52.9, 59, 59.4, 60.5, 60.5 and 51.3 MPa, respectively. The strength values at 60 days were, accordingly, 50.9, 55.5, 63, 63.9, 63.5, 64.9 and 53.5 MPa. It was found that the compressive strength of mortar increased up to 30% with replacement of NS by RIP for all curing ages, which is about 19–30% higher compressive strength relative to the reference mortar for all curing ages, see Figure 4b.

This high strength of mortar can be linked to the higher strength of RIP compared to NS, the morphology of the RIP (e.g., its rough surface), and the angular shape of RIP particles. The rough surface of the RIP may provide the formation of a stronger interfacial transition zone (ITZ) between the cement paste and the aggregate, which is a crucial factor for improving the strength of cement-based composites [32,33]. The findings are in good agreement with the literature [12,13,14,15,16,17,18,19,20]. This study reveals that the optimum replacement content of NS by RIP is 30% due to its significantly higher compressive strength.

#### 3.2.3. Tensile and Flexural Strength

The tensile strength of mortar mixes is presented in Figure 5a and normalized tensile strength is shown in Figure 5b. It can be observed that the tensile strength of mortar increases with up to 30% replacement of NS by RIP for all curing ages compared to the reference mortar (100% NS). The average tensile strengths of mortar tested at 28 days at 0%, 5%, 10%, 15%, 20%, 30% and 50% were 2.68, 2.79, 2.85, 3.02, 3.06, 3.16 and 2.94 MPa, respectively. The increase in the tensile strength of mortar mixes for all curing ages is in the range of 10%–18% of the strength of NS, see Figure 5b. It seems that up to 30% replacement of NS with RIP was beneficial to the tensile strength. Replacement above that number decreased the tensile strength, but not below the tensile strength of the reference mortar. This is in good agreement with the compressive strength results and consistent with the literature [10,12,13,14].

The flexural strength and normalized flexural strength of mortar mixes are presented in Figure 6a. A significant increase in flexural strength was observed with the increased replacement percentage of NS by RIP at all curing ages. Although the maximum flexural strength was observed at 30% RIP, the strength at 50% was still significantly higher than that of the reference mortar. This was consistent with the compressive and tensile strength measurements presented above. The average flexural strength of mortar tested at 28 days at 0%, 5%, 10%, 15%, 20%, 30% and 50% was 4.67, 5.55, 6.29, 6.58, 6.73, 6.88 and 6.29 MPa, respectively. The maximum increase in flexural strength of mortar was observed for 30% RIP, which was about 30–47% higher flexural strength relative to the reference mortar (100% NS mortar), see Figure 6b. As already discussed, in relation to the compressive strength measurements, these higher tensile and flexural strengths of mortar made with RIP could be linked to the higher strength of RIP, higher rough surface, and irregular shape of the RIP, which ensured better bonding between cement paste and RIP than NS.

#### 3.2.4. Dimensional Stability

The dimensional stability of mortar mixtures, measured as length change as a function of time, is given in Figure 7a. Each data point represents the average values of three specimens. The relationships between the maximum length change at 120 days and the replacement percentages of NS are shown in Figure 7b. In general, the overall trend presented in Figure 7a indicates distinct phases in the variation in length change for the different replacement percentages of NS by RIP with time. For the low content of RIP, it was seen that shrinkage occurred for the mortar made with 0% and 5% RIP. The average shrinkage of mortar made with 0% and 5% RIP was −0.044% and −0.04%, respectively, at 28 days, and −0.027% and −0.013%, accordingly, at 120 days. It was noted that the shrinkage of mortar made with 5% RIP decreased (expansion increases) more than that of 0% RIP mortar at a later age. For a moderate content of RIP (10%, 15% and 20%), shrinkage occurred at an early age. Thereafter, an increase in the length change (i.e., expansion) was observed for all mixes (10%, 15% and 20% of RIP). However, the rate of expansion varied for different percentages of RIP.

The change in length of the mixes reached +0.0% at around 28, 8 and 4 days, respectively, for the 10%, 15% and 20% RIP. This trend had already been reported by Maslehuddin et al. [34]. The maximum expansion of the mixes made with 10%, 15% and 20% RIP was +0.016%, +0.026% and +0.03%, respectively, at three months of exposure. It was noted that the expansions of the mixes were below the normally accepted value of +0.05% suggested in ASTM C33 [24] after three months of continuous exposure to a moist environment. The higher expansion of RIP mortar mixes could be attributed to its higher temperature generation in the mixes, as shown in Table 4.

### 3.3. Durability of Mortar

#### 3.3.1. Porosity

In order to understand more comprehensively the role of RIP on the mechanical strength of mortar, the total porosity of concrete was measured by means of water absorption. The results are given in Figure 8a. With the exception of the mortar mix made with 50% RIP, it was confirmed that increasing the replacement percentage of NS by RIP decreased the porosity of mortar. This lower porosity of mortar made with RIP is consistent with the higher mechanical strength of mortar. The apparent average porosity of the mortar made with 0%, 5%, 10%, 15%, 20%, 30% and 50% RIP was 18.40, 16.42, 15.29, 14.03, 12.78, 11.71 and 16.67%, respectively, at 28 days. The maximum decrease in porosity was observed for 30% RIP mortar, which was about 36% lower than the reference mortar (100% NS), see Figure 8b. 

The mechanical strength of quasi-brittle materials is known to be closely linked with porosity [35,36]. The relationship between porosity and strength (compressive, tensile, and flexural) of the mortar mixes is plotted in Figure 9. The measured compressive, tensile, and flexural strength of mortar mixes is compared with models/exponential relationships available in the literature [37,38,39,40,41]. As the porosity increased, the mechanical strength (compressive, tensile, and flexural) decreased, as expected. The relationships between the strength and porosity of the mortar mixes agreed with the relationships presented in the literature (see Figure 9a–c).

Since porosity is one of the key factors influencing the strength of mortar, the following equations were proposed using the best fitted exponential curve to calculate the compressive, tensile, and flexural strength of mortar containing RIP at 28 days: (2)$$f\prime_{c}=100.05e^{-0.039\;*\;P},\;R^{2}=0.8436\;(\mathrm{Compressive}\;\mathrm{strength}\;\mathrm{at}\;28\;\mathrm{days})$$
(3)$$f\prime_{t}=4.0989e^{-0.022\;*\;P},\;R^{2}=0.863\;(\mathrm{Tensile}\;\mathrm{strength}\;\mathrm{at}\;28\;\mathrm{days})$$
(4)$$f\prime_{f}=12.993e^{-0.05\;*\;P},\;R^{2}=0.7502\;(\mathrm{Flexural}\;\mathrm{strength}\;\mathrm{at}\;28\;\mathrm{days})$$
where $f_{c}^{\prime }$ is compressive strength, $f_{t}^{\prime }$ is tensile strength, $f_{f}^{\prime }$ is flexural strength, and *P* is porosity.

#### 3.3.2. Water Absorption

The evolution of the capillary water absorption of mortar is presented in Figure 10a and the normalized absorption is reported in Figure 10b. It can be seen that steep water absorption is observed for all the mortar mixes at up to 2 h of water exposure. Except for 50% RIP, it was noted that as the content of RIP in mortar increased the water absorption decreased. The maximum decrease was observed for the mortar made with 30% RIP, which was about 48% lower than the reference mortar (100% NS) at 120 h. Lower porosity and denser microstructure, including the ITZ, are probably responsible for the lower water absorption for the mortar mixes containing RIP. These results are in good agreement with the porosity measurements. It should be noted that the water absorption increased for the mortar made with 50% RIP, which was consistent with the results of mechanical strength. Although the water absorption capacity was not directly proportional to the mechanical strength, it was interesting to see the role of the RIP on the relationship between these two parameters (Figure 11). It was found that as water absorption increased the mechanical strength (compressive, tensile, and flexural) decreased. This trend is in agreement with the porosity and strength relationship shown in Figure 9. 

The experimental results in the literature showed that water absorption decreased with increasing compressive strength [42]. Three equations are proposed using the best fitted exponential curve to calculate the compressive, tensile, and flexural strength of mortar containing RIP at 28 days:(5)$$f\prime_{c}=81.197e^{-0.066\;*\;WA},\;R^{2}=0.5469\;(\mathrm{Compressive}\;\mathrm{strength}\;\mathrm{at}\;28\;\mathrm{days})$$
(6)$$f\prime_{t}=3.8497e^{-0.048\;*\;WA},\;R^{2}=0.8908\;(\mathrm{Tensile}\;\mathrm{strength}\;\mathrm{at}\;28\;\mathrm{days})$$
(7)$$f\prime_{f}=11.112e^{-0.104\;*\;WA},\;R^{2}=0.7351\;(\mathrm{Flexural}\;\mathrm{strength}\;\mathrm{at}\;28\;\mathrm{days})$$
where $f_{c}^{\prime }$ is compressive strength, $f_{t}^{\prime }$ is tensile strength, $f_{f}^{\prime }$ is flexural strength, and WA is water absorption.

### 3.4. Residual Strength of Mortar After Exposure to High Temperature

The residual compressive strength of the mortar mixes was determined on cube (50 mm) specimens at room temperature after exposure to thermal loads of 120, 250, 400 and 600 °C at a slow heating rate of 2 °C/min. The residual strength of the specimens was compared with the strength measured at ambient temperature (20 °C). Three specimens were used for each temperature and then the average strength was calculated as shown in Figure 12a. For an in-depth analysis of the influence of RIP on the residual compressive strength of mortar, it is worthwhile to study the variation in strength, normalized with respect to the reference compressive strength tested at 20 °C, i.e., $f_{c}^{,T}$/ $f_{c}^{,20^{o}C}$. As expected, as the temperature increased the residual compressive strength of mortar decreased. The strength decreased at 120 °C for mortar of about 5–9%, for all mortar mixes. The reduction of strength at this temperature could be due to increasing pore volume caused by the removal of the free water and the physically bound water, which increases the porosity as well as the permeability of mortar [31,43]. At 250 °C, a significant reduction in compressive strength was observed, compared to ambient temperature (20 °C). The situation was worse for the mortar made with RIP. Here, the strength was reduced by 20%, 19%, 25%, 30%, 33%, 30% and 34%, respectively, compared to the original strength at 20 °C for mortar made with 0%, 5%, 10%, 15%, 20%, 30% and 50% RIP.

At this temperature, the vapor pore pressure could play an important role due to the dense microstructure of the matrix (higher mechanical strength due to lower porosity), which could produce significantly higher steam pressure, followed by higher tensile stress, resulting in more cracking (higher permeability) and lower strength. The evidence of significantly higher vapor pore pressure of 4.5 MPa was observed at around 250 °C [44]. It was noted that the strength of mortars made with RIP at 20 °C was higher than the reference mortar, which could prevent the escape of steam, due to the denser microstructure. However, at high temperatures, only 54 to 64 percent for 400 °C and 34 to 48 percent for 600 °C of the original strength was left. This reduction increased with the increasing percentage of RIP. This means that the mortar significantly degraded, from a mechanical point of view. At these temperatures, the release of absorbed and chemically bound water, cement dehydration (decomposition of calcium hydroxide and C–S–H phases), and the coefficient of thermal expansion (the expansion could be higher for RIP than NS due to possible higher thermal conductivity of RIP than NS) took place, which can cause micro-cracks in the matrix of the mortar [45].

It was noted that a significantly higher decay of mechanical strength was observed for the mortar made with a higher percentage of RIP compared to the reference, and it was more pronounced at high temperatures. At high temperatures, a significant increase in permeability may have occured due to the development of cracks mainly caused by the thermal incompatibility between the cement paste and the aggregates [45]. In general, at high temperatures, the cement paste shrinks due to dehydration while the aggregates (coarse and fine) dilate due to their thermal expansion [31,45]. This mismatch induces tensile stresses in the matrix and compressive stresses in the aggregates, which in turn can lead to crack formation [31]. It is believed that mortar made with RIP is more thermally conductive, and that its thermal conductivity increases with the increasing percentage of RIP. Therefore, the mismatch between cement paste and RIP could be higher (due to higher expansion of iron material) than the NS, which will induce more cracking. The evidence of more cracking in the RIP mortar exposed to 600 °C can be clearly seen in Figure 13b. This could explain the lower strength of mortars with RIP after exposure to high temperatures.

## 4. Conclusions

In this paper, the effect of the incorporation of recycled iron powder (RIP) as a partial replacement for natural sand (NS) in mortar was investigated to quantify the mechanical strength (compressive, tensile, and flexural), durability, and decay of mechanical strength after exposure to high temperatures (20, 120, 250, 400 and 600 °C). It is worth noting that, while some studies have been conducted on the mechanical properties of concrete containing RIP, limited information is available on mortars with RIP, especially those related to shrinkage, durability, and exposure to high temperature. Within this context, seven mortar mixes made with different percentage replacement of NS by RIP were studied. The main findings of the influence of RIP on the mechanical, durability, and residual strength can be summarized as follows:(1)The workability of mortar decreased with the increasing percentage replacement of NS by RIP due to the highly angular and rough surface of RIP, which provided better interlocking, resulting in restraint to the mobility of fresh mortar.(2)The mechanical strength of mortar increased with up to 30% replacement of NS by RIP, which was 30% for compressive, 18% for tensile, and 47% for flexural, respectively—higher than that of reference mortar (100% NS mortar) at 28 days. This higher mechanical strength could be attributed to the higher strength of RIP, its rougher surface, and the more angular shape of RIP particles compared to NS, which ensures better bonding between cement paste and RIP compared to NS.(3)Shrinkage was observed at lower concentrations of RIP (0% and 5%), and shrinkage and expansion were found at moderate concentrations of RIP (10%, 15%, and 20%). Finally, only expansion was observed for the mortar with higher RIP content (30% and 50%).(4)Except for 50% RIP, the porosity and water absorption decreased with the increased replacement percentage of NS by RIP, due to better ITZ and the denser microstructure of mortar made with RIP. However, a satisfactory relationship between mechanical strength (compressive, tensile, and flexural) and durability properties (porosity and water absorption) was observed for mortars incorporating RIP, which was consistent with the literature.(5)As temperature increased, the decline in the mechanical strength of the mortar increased as well. The higher the temperature the higher the decrease. The decrease in strength was higher for the mortar made with RIP than with NS. This could be linked to the release of absorbed and chemically bound water, the decomposition of calcium hydroxide and C–S–H phases, and the coefficient of thermal expansion, which can cause micro-cracks in the matrix of the mortar, resulting in lower strength.
From the experimental results of the seven mortar mixes, this study revealed that RIP can be used as a replacement for up to 30% NS due to its higher mechanical strength and better durability. The outcome of this research work will encourage iron waste producers to continue collecting and storing these hazardous materials to be used as fine aggregates in concrete construction.

## Figures and Tables

**Figure 1 materials-13-01168-f001:**
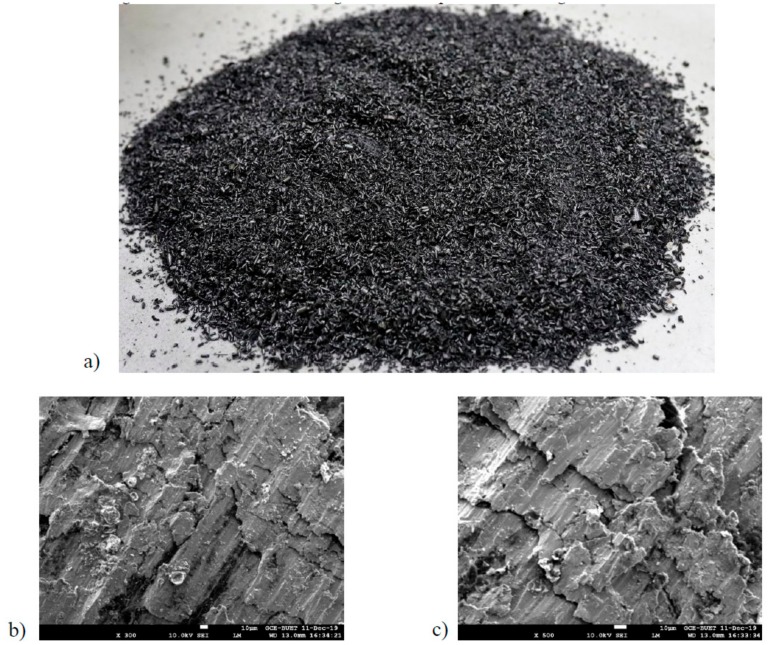
The recycled iron powder used as a fine aggregate (**a**) and the SEM images of recycled iron powder (**b**–**c**).

**Figure 2 materials-13-01168-f002:**
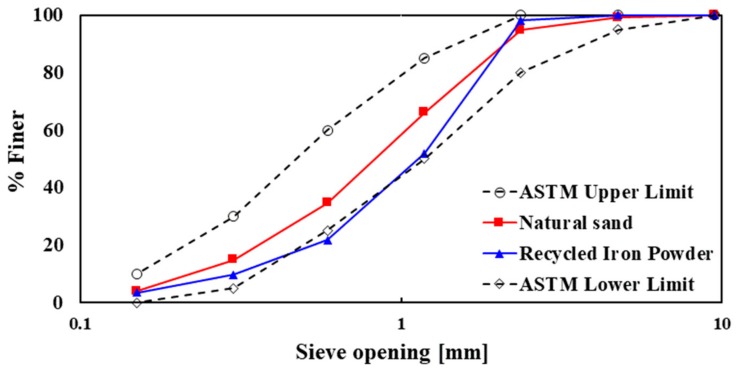
Grading curve of natural sand (NS) and recycled iron powder (RIP), and comparison with the upper and lower limit recommended in the ASTM C33 standard [24].

**Figure 3 materials-13-01168-f003:**
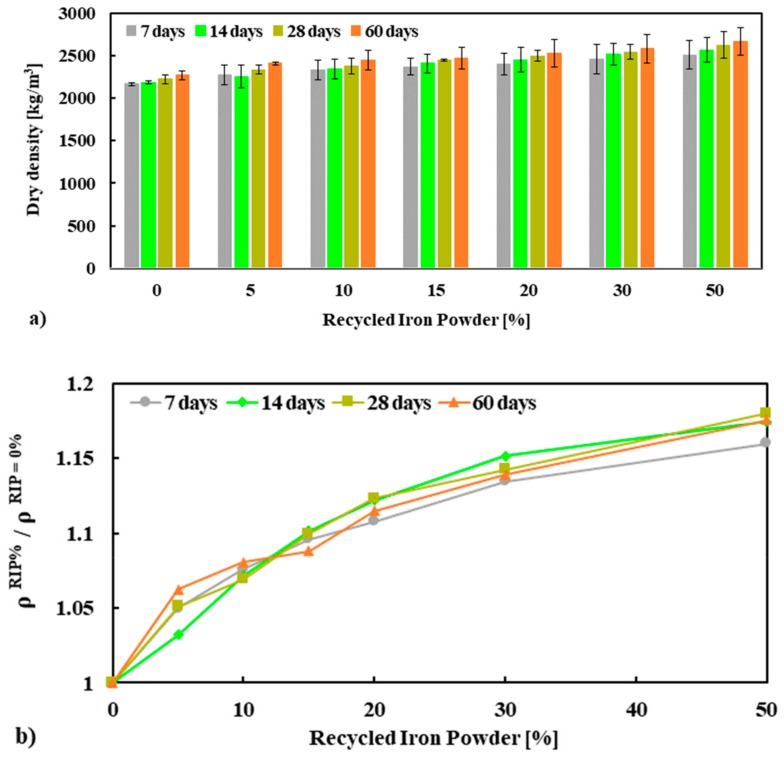
Dry density (ρ) (**a**) and normalized ρ (**b**) of mortars as a function of RIP content conducted at 7, 14, 28 and 60 days.

**Figure 4 materials-13-01168-f004:**
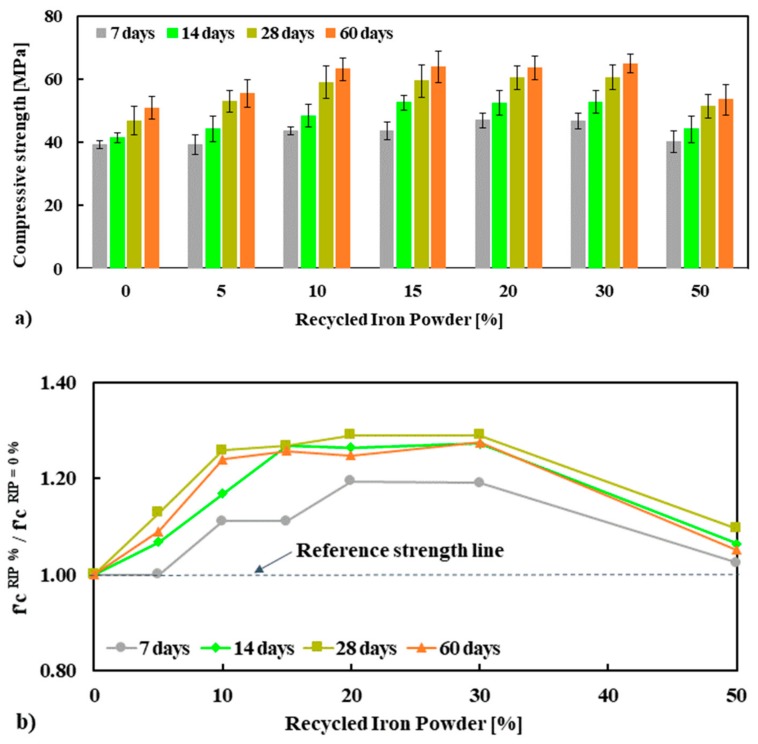
Compressive strength (f′c) (**a**) and normalized f′c (**b**) of mortar specimens tested at 7, 14, 28 and 60 days.

**Figure 5 materials-13-01168-f005:**
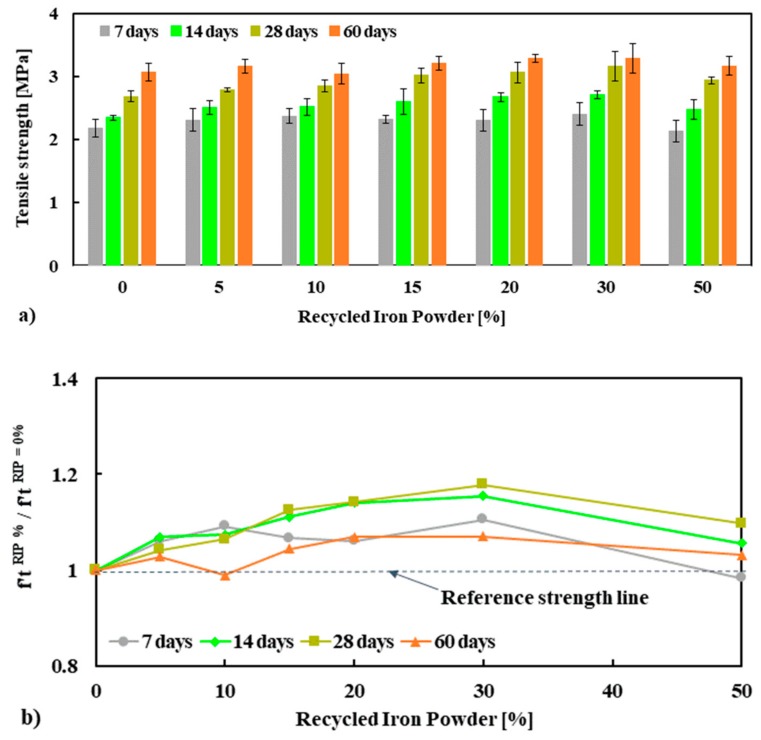
Tensile strength (f′t) (**a**) and normalized f′t (**b**) of mortar specimens conducted at 7, 14, 28 and 60 days.

**Figure 6 materials-13-01168-f006:**
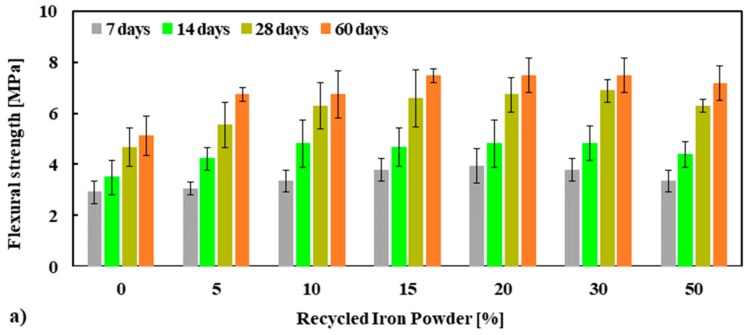
Flexural strength (f′f) (**a**) and normalized f′f (**b**) of mortars specimens performed at 7, 14, 28 and 60 days.

**Figure 7 materials-13-01168-f007:**
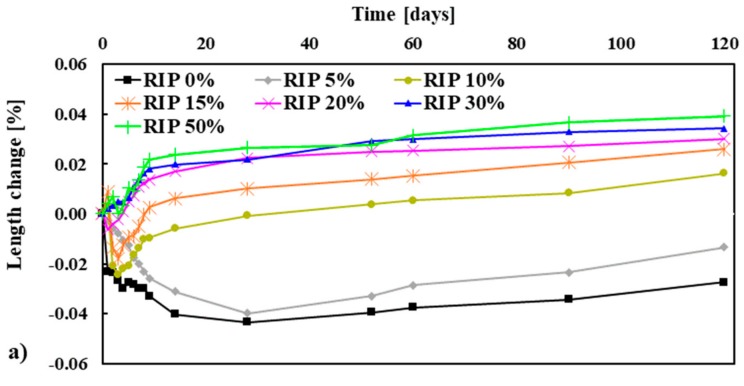
Length change (**a**) and maximum length change measured at 120 days (**b**) of mortar mixes.

**Figure 8 materials-13-01168-f008:**
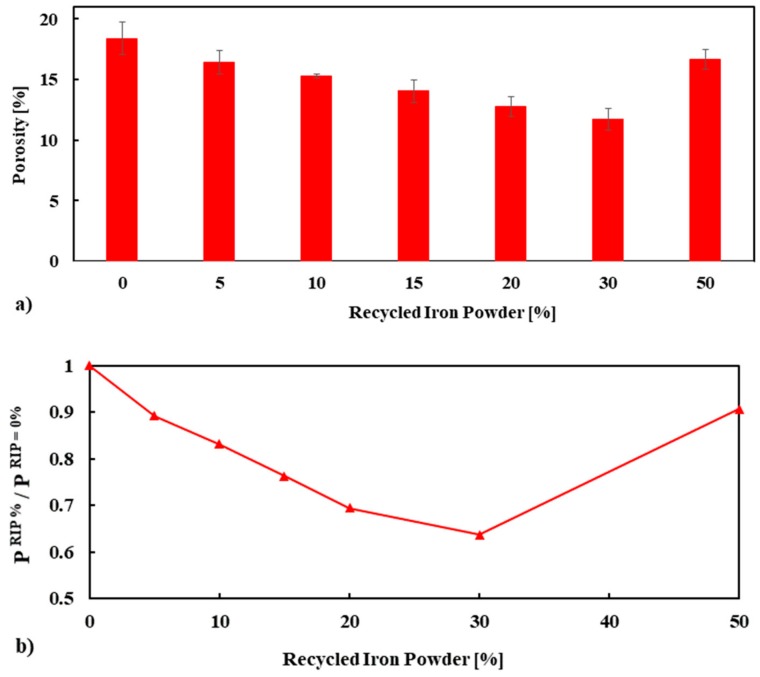
Apparent porosity (P) (**a**) and normalized P (**b**) of mortar specimens performed at 28 days.

**Figure 9 materials-13-01168-f009:**
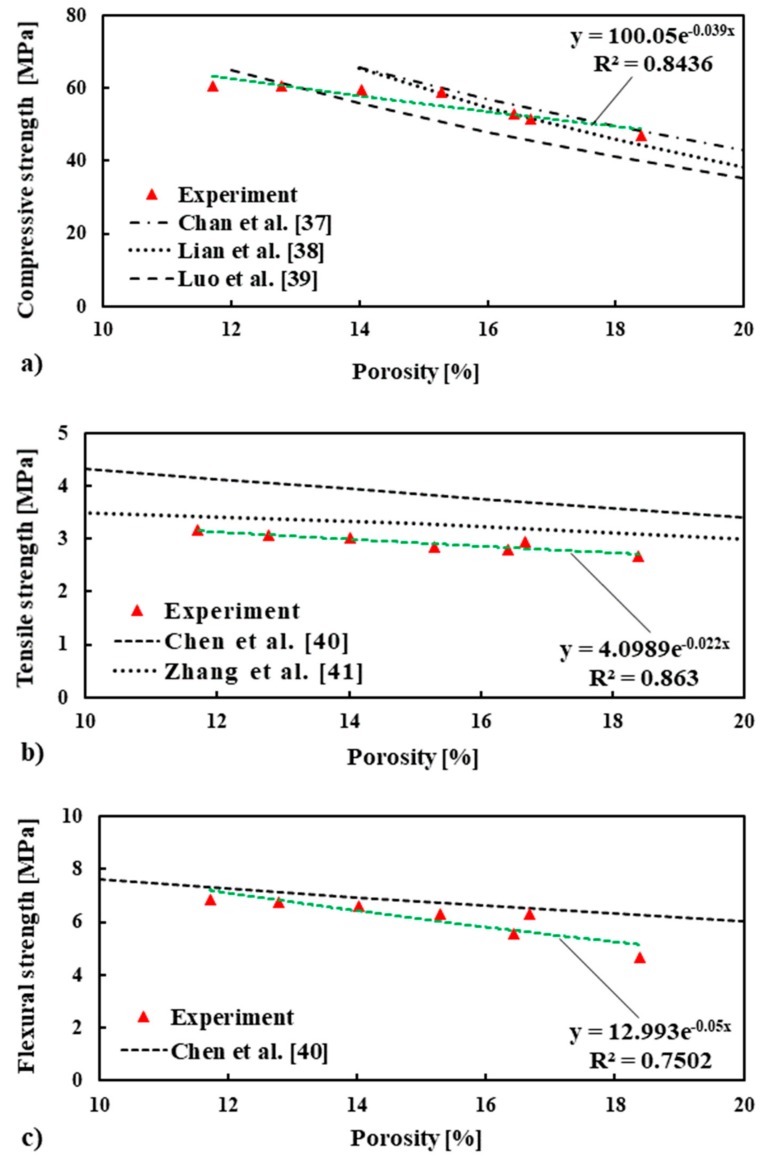
The relations of apparent porosity with compressive (**a**), tensile (**b**), and flexural strength (**c**), respectively, are compared with different models/exponential relationships available in the literature.

**Figure 10 materials-13-01168-f010:**
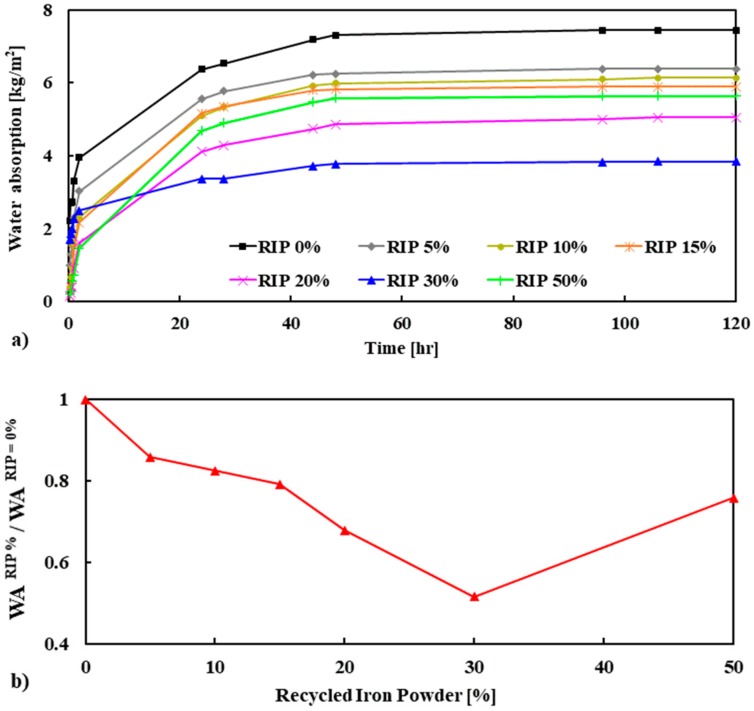
Capillary water absorption (WA) (**a**) and normalized WA (**b**) of mortar specimens.

**Figure 11 materials-13-01168-f011:**
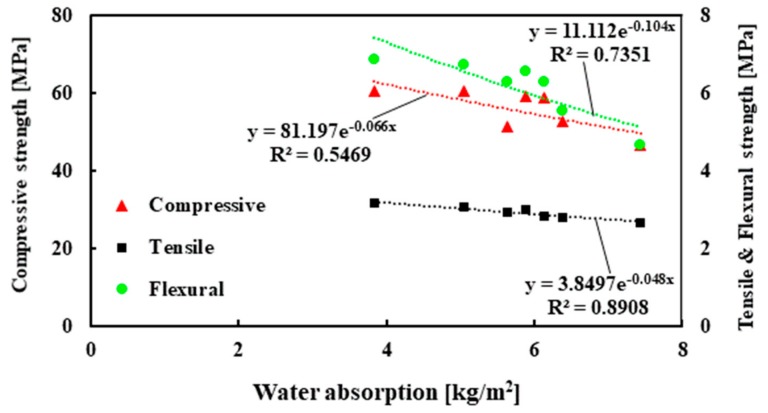
The relation of the water absorption with compressive, tensile, and flexural strength of the mortar.

**Figure 12 materials-13-01168-f012:**
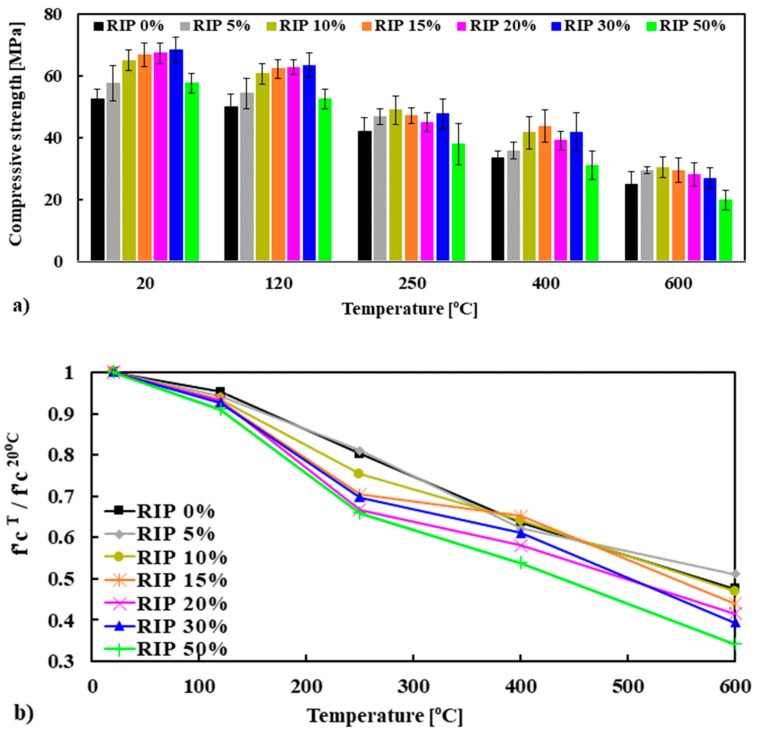
Residual (f’c) (**a**) and normalized (f’c) (**b**) compressive strength of mortar at high temperatures.

**Figure 13 materials-13-01168-f013:**
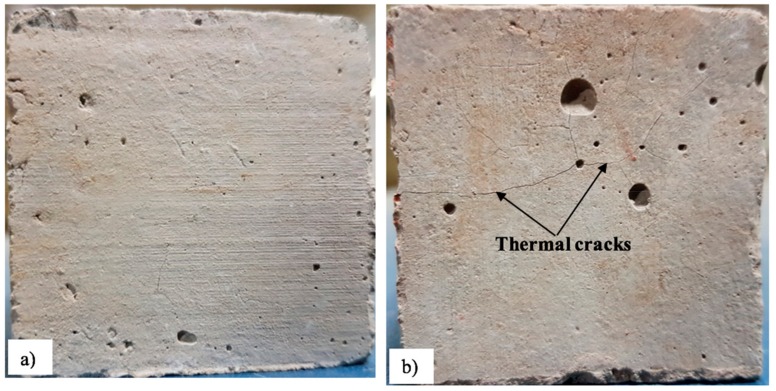
Images of mortars specimens subjected to 600 °C: (**a**) reference mortar (0% RIP) and (**b**) mortar made with 50% RIP.

**Table 1 materials-13-01168-t001:** Physical properties of fine aggregates.

ID	Fineness Modulus	Specific Gravity	Absorption Capacity [%]
NS	2.86	2.56	5.9
RIP	3.15	4.3	2.6

**Table 2 materials-13-01168-t002:** Chemical compositions of cement and recycled iron powder.

Chemical Composition	Cement [%]	RIP [%]
SiO_2_	24.90	8.46
Fe_2_O_3_	3.96	87.46
Al_2_O_3_	7.52	0.87
K_2_O	1.00	0.28
CaO	53.43	1.08
TiO_2_	1.18	0.08
MgO	2.52	0.32
Na_2_O	0.27	0.27
SO_3_	4.77	0.88
P_2_O_5_	0.21	0.17
Cr_2_O_3_	0.08	0.13

**Table 3 materials-13-01168-t003:** Mixture proportion of mortar mixes.

Mix ID	Cement [kg/m^3^]	Fine Aggregates [kg/m^3^]		Water [kg/m^3^]
		Sand [kg/m^3^]	RIP [kg/m^3^]	
RIP 0%	810.9	1216.4	0.0	243.3
RIP 5%	810.9	1155.5	103.2	243.3
RIP 10%	810.9	1094.7	206.5	243.3
RIP 15%	810.9	1033.9	309.7	243.3
RIP 20%	810.9	973.1	413.0	243.3
RIP 30%	810.9	851.5	619.5	243.3
RIP 50%	810.9	608.2	1032.5	243.3

**Table 4 materials-13-01168-t004:** Slump and temperature of fresh mortar mixes.

RIP	0%	5%	10%	15%	20%	30%	50%
Slump [cm]	16	15.5	14	14.5	13	12	9.5
Temperature [°C]	22	23	24	25.5	26	26.5	28

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
