# Peer review of "Effect of Recycled Iron Powder as Fine Aggregate on the Mechanical, Durability, and High Temperature Behavior of Mortars"

_materials, 2020, doi:10.3390/ma13051168_

Round 1
Reviewer 1 Report
There is no doubt that the topic is of great interests to engineering practice.
The objective of the paper is to evaluates the mechanical, durability, and residual compressive strength of sustainable mortar that uses recycled iron powder as fine aggregates. It was found that the mechanical strength of mortar increases up to 30% replacement of NS by RIP, and the increase is 30% for compressive, 18% for tensile, and 47% for flexural strength at 28 days, respectively, compared to the reference mortar. The study demonstrates that up to 30% of RIP can be utilized as fine aggregates in the mortar, which results in the conservation of natural sand, and produces an energy-efficient sustainable building material.
The subject of the paper is interesting, but the authors should consider "major comprehensive" revisions to improve the manuscript. Please see remarks below.
(1) Please reorganize and develop the references in the introduction section to make it more coherent and logical about this specific topic described in the paper.
(2) Could you please extend your research with the statistical analysis as it is important from the point of conclusion presented in the paper.
(3) Line 104 editor correction is needed.
(4) Figure 8a - mistake in the title
(5) Conclusions have to be more explored covering issue marked in point (1)
Author Response
The response to reviewer 1 is given in the attached file.

Reviewer 2 Report
-The words sustainable or environmentally friendly must be removed. The study has not evaluated the LCA of the material
-Reference 1 is not the most adequate because does not related to a study on aggregate availability
-Also saying that some problems have not enough aggregates is not a suitable approach. The authors must instead mention the importance of the circular economy. See for instance
COM (2014) 398 Final. Towards a circular economy: A zero waste programme for Europe. European Commission, Brussels
EC (European Commission), 2015, Closing the Loop – An EU Action Plan for the Circular Economy, Brussels
-They should cite recent review works about recycling in the built environment:
Pacheco-Torgal, F.; Ding, Y.; Koutamanis, A.; Colangelo, F.; Tuladhar, R.. eds. 2019. Advances in construction and demolition waste recycling: Management, processing and environmental assessment" ed. 2. Abington Hall, Cambridge: Elsevier Science and Technology
Mahpour, A. (2018). Prioritizing barriers to adopt circular economy in construction and demolition waste management. Resources, Conservation and Recycling, 134, 216-227.
-Line 57: “This byproduct is generated in enormous quantities”. Provide data
-Lines 58-59: “RIP can be hazardous to human health since it can be easily inhaled”. Provide references
Author Response
The response to reviewer 2 is given in the attached file.

Round 2
Reviewer 1 Report
Remarks from previous review were corrected.